# F-region drift current and magnetic perturbation distribution by X wave heating ionosphere

Yong Li [1,2], Hui Li[2], Jian Wu[1,2], Xingbao Lv [1,3,4], Chengxun Yuan[1,3,4] , Ce Li[1], Zhongxiang Zhou[1,3,4]

[1] School of Physics, Harbin Institute of Technology, Harbin 150001, China
[2] China Research Institute of Radio Wave Propagation, Beijing 102206, China
[3] Heilongjiang Provincial Key Laboratory of Plasma Physics and Application Technology, Harbin 150001, China
[4] Heilongjiang Provincial Innovation Research Center for Plasma Physics and Application Technology, Harbin 150001, China

*Correspondence to*: Hui Li( lihui_2253@163.com); Zhongxiang Zhou(zhouzx@hit.edu.cn)

**Abstract.** We present a theoretical and numerical study of the drift current and magnetic perturbation model in the ionosphere by incorporating the ohmic heating model and the magnetohydrodynamic (MHD) momentum equation. Based on these equations, the ionospheric electron temperature and drift current are investigated. The results indicate that the maximum change of electron temperature $\Delta T_\text{e}$ is about 570 K, and the ratio is $\Delta T_e / T_e \sim 48\%$ . The maximum drift current density is $8 \times 10^{-10}\,\text{A} \cdot m^{-2}$, and its surface integral is 5.76 A. Diamagnetic drift current is the main form of current. The low collision frequency between charged particles and neutral particles has little effect on the current, and the collision frequency of electrons and ions is independent of the drift current. The current density profile is a flow ring. We present the effective conductivity as a function of the angle between the geomagnetic field and the radio wave; the model explains why the radiation efficiency was strongest when the X wave is heating along the magnetic dip angle as reported in recent observations by Kotik et al. We calculate the magnetic field variation in the heating region based on the MHD theory; the results show that the maximum magnetic field perturbation in the heating area is 48 pT.

## 1 Introduction

Extremely low frequency (ELF) waves have irreplaceable advantages in communication, navigation, and magnetospheric studies. In the 1970s, Willis and Davis first proposed the theory of modulating the ionosphere to excite ELF waves(Willis and Davis, 1973). Then Getmantsev et al. successfully excited ELF signals in experiments(Getmantsev et al., 1974).

There are several main physical mechanisms of ELF signal excitation by heating the ionosphere. The first mechanism is called the polar electrojet model (PEJ). A polar electrojet is a strong horizontal electric current driven by an atmospheric dynamo electric field and a magnetospheric electric field. It can be effectively modulated by heating the ionosphere with a modulated high-frequency (HF) wave. The resulting modification of the electrojet current creates an effective antenna radiating at the modulation frequency(Stubbe et al., 1981; Stubbe and Kopka, 1977; Rietveld et al., 1987). Numerous researchers have analyzed this process theoretically and experimentally, and proposed optimization measures such as preheating(Milikh and Papadopoulos, 2007), geometric modulation(Cohen et al., 2008, 2009), and beam painting(Papadopoulos et al., 1990) to enhance the radiation signal. The shortcoming of PEJ is that the electric field changes suddenly and is difficult to predict(Belyaev et al., 1987). The

second mechanism is beat-wave (BW) modulation(Yang et al., 2019). BW modulation can excite ELF waves by dividing the heating source into two groups(Ganguly, 1986; Kuo et al., 2012), in which one group transmits a continuous wave at a frequency $f_0$, and the other group transmits a continuous wave at a frequency $f_0 \pm f$ ( $f$ is the ELF/VLF modulation frequency). Barr and Stubbe utilized this mechanism to excite 565 and 2005 Hz signals at Tromsø(Barr and Stubbe, 1997). They thought the BW mode could be approximately equivalent to the beat-wave AM mode, which may be affected by the natural current intensity. However, Kuo et al. proposed that BW modulation can excite another electrojet-independent ELF/VLF signal, which is driven by the ponderomotive force.(Kuo et al., 2010; Kuo et al., 2012; Kuo et al., 2011)

In 2011, Papadopoulos proposed an ionospheric current drive (ICD) model based on the experimental results of the High Frequency Active Auroral Research Program (HAARP)(Papadopoulos et al., 2011a). Papadopoulos proposed that ELF current is driven in a two-step process based on the model of Lysak(Papadopoulos et al., 2011b; Lysak, 1997). The idea is that HF heating creates a pressure gradient in the heated region, then leading to a diamagnetic current that excites a hydromagnetic wave with the modulation frequency. Kotik et al. verified the mechanism experimentally in SURA. They discussed the effects of HF emission frequency, emission direction, and magnetic field activity on radiation signals(Kotik et al., 2015; Kotik et al., 2013). Eliasson et al. established the propagation model of ELF waves in the polar region based on the Hall MHD model(Eliasson et al., 2012). Sharma extended the radiation propagation model in the mid and low latitudes (Sharma et al., 2016). In 2019, Mahmoudian demonstrated that the VLF signal may not penetrate the D-region as efficient as the ELF signal(Mahmoudian and Kalaee, 2019).

At present, theoretical research on ICD theory focuses mainly on the propagation process of ELF wave. In this paper, considering the effect of transmitter parameters and ionospheric parameters, we develop the ionospheric drift current and magnetic perturbation model by coupling the ohmic heating and MHD momentum equations. We then study the drift-current properties and the effects of collisions and transmitter angle on the drift current, and we calculate the magnetic field variation in the heating region.

This paper is organized as follows: In Section 2, we give the ohmic heating model for tensor conductivity and derive a formula for ionospheric drift current using the MHD momentum equation. In Section 3, numerical solutions of the model are presented for realistic ionospheric profiles, drift current properties are discussed, and the effect of emission angle is analyzed. Finally, in Section 4, the conclusions are presented.

## 2 Theoretical model

### 2.1 HF heating model

Background ionospheric data used in this work are obtained from HAARP(magnetic inclination 75°). Referring to previous literature(Papadopoulos et al., 2011b), the magnetic field inclination is assumed to be $90^\circ$. The heating model is simplified into

a two-dimensional plane, in which the z-axis is parallel to the geomagnetic field, and the x-axis is perpendicular to the magnetic field. The ohmic heating equation is (Shoucri et al., 1984; Lofas et al., 2009)

$$\frac{3}{2}k_B N_e \frac{\partial T_e}{\partial t} = \nabla \cdot (\overline{\overline{K_e}} \cdot \nabla T_e) + Q_{HF} + Q_0 - L_e(T_e, T_0) \tag{1}$$

where $k_B$ is Boltzmann's constant, $N_e$ is electron density, $T_e$ is electron temperature, $Q_0$ is the background power source, $Q_{HF}$ is ohmic heating by high-power radio waves, $L_e(T_e, T_0)$ is the rate of energy loss due to both elastic and inelastic collisions with ions and neutral particles, and $\overline{\overline{K_e}}$ is the thermal conductivity tensor, which comes from Banks(Kockarts, 1973)

$$\overline{\overline{K_e}} = \begin{pmatrix} 0 & 0 & 0 \\ 0 & 0 & 0 \\ 0 & 0 & K_{e0} \end{pmatrix} \tag{2}$$

$$K_{e0} = \frac{7.7 \times 10^5 T_e^{5/2}}{1 + 3.22 \times 10^4 (T_e^2 / N_e) \sum_n N_n Q_D} \tag{3}$$

where $N_n$ is the density of neutral particles of species *n*, and $Q_D$ is the average momentum transfer cross-section, which is calculated by Schunk and Nagy(Schunk and Nagy, 2009). $Q_{HF}$ is calculated from Joule heating,

$$Q_{HF} = \frac{1}{2} \text{Re}\left[ \mathbf{E}^* \cdot \overline{\overline{\sigma}} \cdot \mathbf{E} \right] \tag{4}$$

where $\overline{\overline{\sigma}}$ is the conductivity tensor(Gurevich, 2012)

$$\overline{\overline{\sigma}} = \begin{pmatrix} \sigma_{xx} & \sigma_{xy} & \sigma_{xz} \\ \sigma_{yx} & \sigma_{yy} & \sigma_{yz} \\ \sigma_{zx} & \sigma_{zy} & \sigma_{zz} \end{pmatrix} \tag{5}$$

$$\begin{aligned} \sigma_{xx} = \sigma_{yy} &= \frac{\varepsilon_0 \omega_{pe}^2 v_e}{2}\left[ \frac{1}{(\omega - \omega_{ce})^2 + v_e^2} + \frac{1}{(\omega + \omega_{ce})^2 + v_e^2} \right] \\ \sigma_{xy} = -\sigma_{yx} &= -i\frac{\varepsilon_0 \omega_{pe}^2 v_e}{2}\left[ \frac{1}{(\omega - \omega_{ce})^2 + v_e^2} - \frac{1}{(\omega + \omega_{ce})^2 + v_e^2} \right] \\ \sigma_{zz} &= \frac{\varepsilon_0 \omega_{pe}^2 v_e}{\omega^2 + v_e^2} \\ \sigma_{xz} = \sigma_{zx} &= \sigma_{yz} = \sigma_{zy} = 0 \end{aligned} \tag{6}$$

where $\varepsilon_0$ is the vacuum dielectric constant; $\omega, \omega_{pe}, \omega_{pe}, \nu_e$ are, respectively, the frequency of the incident wave, the ionospheric frequency, the cyclotron frequency, and the frequency of electron collision with other particles; $\mathbf{E}$ is the incident electric field; and the incident wave is generally an X wave or O wave.

$$\mathbf{E}_X = E_0(s)\sin(\theta)x + iE_0(s)y + E_0(s)\cos(\theta)\hat{z} \qquad \text{X wave}$$
$$\mathbf{E}_O = E_0(s)\sin(\theta)x - iE_0(s)y + E_0(s)\cos(\theta)\hat{z} \qquad \text{O wave}$$

(7)

where $\theta$ is the angle between the incident wave and the z-axis and $E_0(s)$ is the electric field intensity in the ionosphere(Gustavsson et al., 2010)

$$E_0(s) = E(s_0)\left(\frac{s_0}{s}\right)\left[\frac{\varepsilon(s_0)}{\varepsilon(s)}\right]^{0.25}\exp\left(ik_0\int_{s_0}^{s}N(s)ds\right)$$

(8)

where s is the coordinate along the propagation direction of the wave, $\varepsilon(s)$ is the relative dielectric constant, $N(s)$ is the refractive index of the wave in the ionosphere, and the electric field amplitude $E(s_0)$ is estimated by an empirical formula,

$$E(s_0) = \frac{\sqrt{30P_{ER}}}{s_0}$$

(9)

where $P_{ER}$ is the effective radiated power of the transmitter. $L_e(T_e, T_0)$ is the electron cooling rate, which depends mainly on the elastic electron-ion collisions, the elastic electron–neutral collisions, the rotational and vibrational excitation of $N_2$ and $O_2$, and the fine structure excitation of O(Moore, 2007).

**2.2 Ionospheric drift current model**

In this paper, we think there is no drift current in the magnetic field direction because of the ionospheric electric neutrality (Chen, 2012). We consider mainly the current induced perpendicular to the magnetic field and ignore the current parallel to the field. In order to simplify the calculation, the positive ion is set as a single $O^+$ ion, and the collision between electrons and ions $\nu_{ei}$, electrons and neutral particles $\nu_{en}$, and ions and neutral particle $\nu_{in}$ are considered. The influence of neutral wind is ignored. The momentum equation can be written.

$$m\frac{d\mathbf{V}_{e\perp}}{dt} = -e\mathbf{E} - e\mathbf{V}_{e\perp}\times\mathbf{B} - \frac{\nabla_\perp P_e}{N_e} - m\nu_{en}\mathbf{V}_{e\perp} - m\nu_{ei}(\mathbf{V}_{e\perp} - \mathbf{V}_{i\perp})$$

(10)

$$M\frac{d\mathbf{V}_{i\perp}}{dt} = e\mathbf{E} + e\mathbf{V}_{i\perp}\times\mathbf{B} - M\nu_{in}\mathbf{V}_{i\perp} + m\nu_{ei}(\mathbf{V}_{e\perp} - \mathbf{V}_{i\perp})$$

(11)

In this work, we focus on the steady state. Therefore, the left sides of Eqs. (10) and (11) are ignored. The electric force can also

be ignored since this paper focuses on low frequency. The current can be expressed as

$$J_{\perp} = N_e e(\mathbf{V}_{i\perp} - \mathbf{V}_{e\perp})  \tag{12}$$

Solving Eq. (10), (11), and (12), we get

$$J_x = e \frac{\nabla_{\perp} P_e \left( m \nu_{en} + M \nu_{in} \right) \left( M \nu_{ei} \nu_{in} + m \omega_{ce}^2 \right)}{m \left( \nu_{ei}^2 \left( m \nu_{en} + M \nu_{in} \right)^2 + m \left( m \nu_{en} \left( 2\nu_{ei} + \nu_{en} \right) + 2 M \nu_{ei} \nu_{in} \right) \omega_{ce}^2 + m^2 \omega_{ce}^4 \right)}  \tag{13}$$

$$J_y = -e \frac{\nabla_{\perp} P_e \omega_{ce} \left( M (\nu_{ei} - \nu_{en}) \nu_{in} + m(\nu_{in} \nu_{en} + \omega_{ce}^2) \right)}{\nu_{ei}^2 (m \nu_{en} + M \nu_{in})^2 + m(m \nu_{en} (2\nu_{ei} + \nu_{en}) + 2 M \nu_{ei} \nu_{in}) \omega_{ce}^2 + m^2 \omega_{ce}^4}  \tag{14}$$

where

$$\nu_{en} = 2.33 \times 10^{-17} N_{N_2} (1 - 1.21 \times 10^{-4} T_e) T_e + 2.65 \times 10^{-16} N_O T_e^{0.5}$$
$$+ 1.82 \times 10^{-16} N_{O_2} (1 + 0.036 T_e^{0.5}) T_e^{0.5}  \tag{15}$$

$$\nu_{ei} = 5.4 \times 10^{-5} N_e / T_e^{1.5}  \tag{16}$$

$$\nu_{in} = 6.64 \times 10^{-16} n(O_2) + 3.67 \times 10^{-17} n(O) T_i^{0.5} [1 - 0.064 \log 10(T_i)]^2$$
$$+ 6.82 \times 10^{-16} n(N_2)  \tag{17}$$

The spatial distribution of electron pressure can be obtained by coupling with the ohmic heating model of the ionosphere. The spatial distribution of drift current can then be obtained by substituting pressure into Eq. (13) and (14).

**3 Simulation results and discussion**

In this section, we analyze the drift current caused by ohmic heating according to the theoretical model developed in the preceding section. Background data are from HAARP on 2 October 2011. The ionospheric and atmospheric background profiles are given by the International Reference Ionosphere(IRI) model(Bilitza et al., 2017) and the neutral atmosphere model (MSIS)(Picone et al., 2002), as well as geomagnetic field data from the International Geomagnetic Reference Field(IGRF) model(Finlay et al., 2010). Figure 1 shows the background data. The critical frequency of the ionosphere is 3.67 MHz and its altitude is 350 km.

The computational domain is –150 to 300 km in the x-axis direction, and 150 to 450 km in the z-axis direction. The spatial grid size is 2 km. The ERP of the transmitter is set at 500 MW; the transmitting frequency is set at 4 MHz, which is greater than the ionospheric critical frequency; the transmitting half-width of the transmitter is set at 7°; and the transmitting waveform is an X wave.

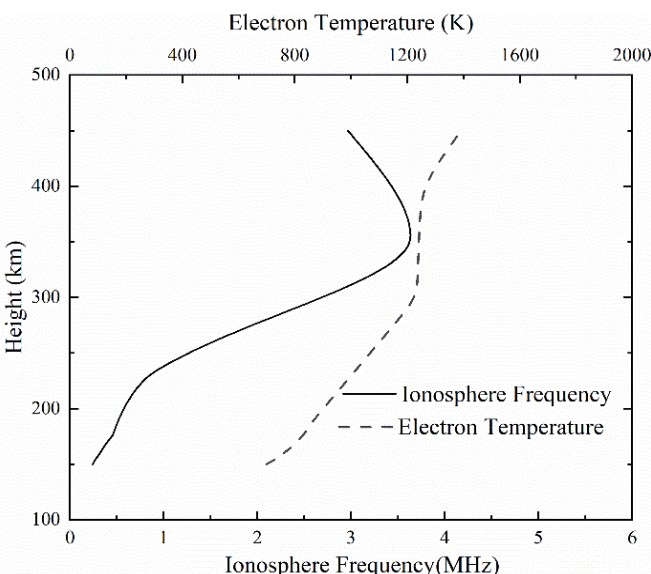

**Figure 1. Background ionospheric electron frequency and electron temperature.**

### 3.1 Ionospheric heating effect and drift current

Based on the presented theory and parameters, we calculate the ionospheric heating results at $\theta = 0$. Figure 2 shows the change in the ionosphere after the heating is stable. Figure 2(a) shows that the maximum temperature change is $\Delta T_e \sim 570$ K when heating is stable, and the change ratio is $\Delta T_e / T_e \sim 48\%$. The corresponding electron pressure is given in Fig. 2(b); it is about $4.1 \times 10^{-9}$ Pa in the center of the heated area. According to the pressure changes obtained from Fig. 2(b), the ionosphere's current density distribution can be obtained by Eq. (15) and (16). The results are shown in Fig.2(c) and (d), the maximum value of $J_y$ is approximately $7.8 \times 10^{-10}$ A·m$^{-2}$, and $J_x$ is approximately $9.3 \times 10^{-43}$ A·m$^{-2}$. The $J_y$ direction is perpendicular to the magnetic field, and $J_x$ is along the pressure gradient.

To characterize the impact of collisions on the current, we calculate various collision frequencies at the position of ionospheric critical frequency and find $v_{en} = 2$ Hz, $v_{in} = 0.29$ Hz, $\omega_{ce} = 1.28$ MHz. Therefore, ignoring the collision frequency of electrons and ions with neutral particles in Eq. (13) and (14) is reasonable, and the equations can be solved to give

$$J_x \approx 0, J_y \approx -e\nabla_\perp P_e / m\omega_{ce} \tag{18}$$

We can find that no current is generated in the x direction, and the current generated in the y direction is mainly a diamagnetic drift current. What is interesting about this simplification is that we don't constrain the electron-ion collisions, so the electron-

ion collisions do not affect the F layer drift current. When $J_y$ is positive, the current flows inward perpendicular to the xz plane; when it is negative, the current flows outward. Therefore, the diamagnetic current is cylindrically symmetric about the z axis.

The distribution of current in the horizontal plane at the critical frequency position is shown in Fig. 3 (obtained by sweeping). The arrow in the figure indicates that the direction of current flow is counterclockwise in this frame, with zero current in the heating center, gradually increasing and then decreasing towards the outside.

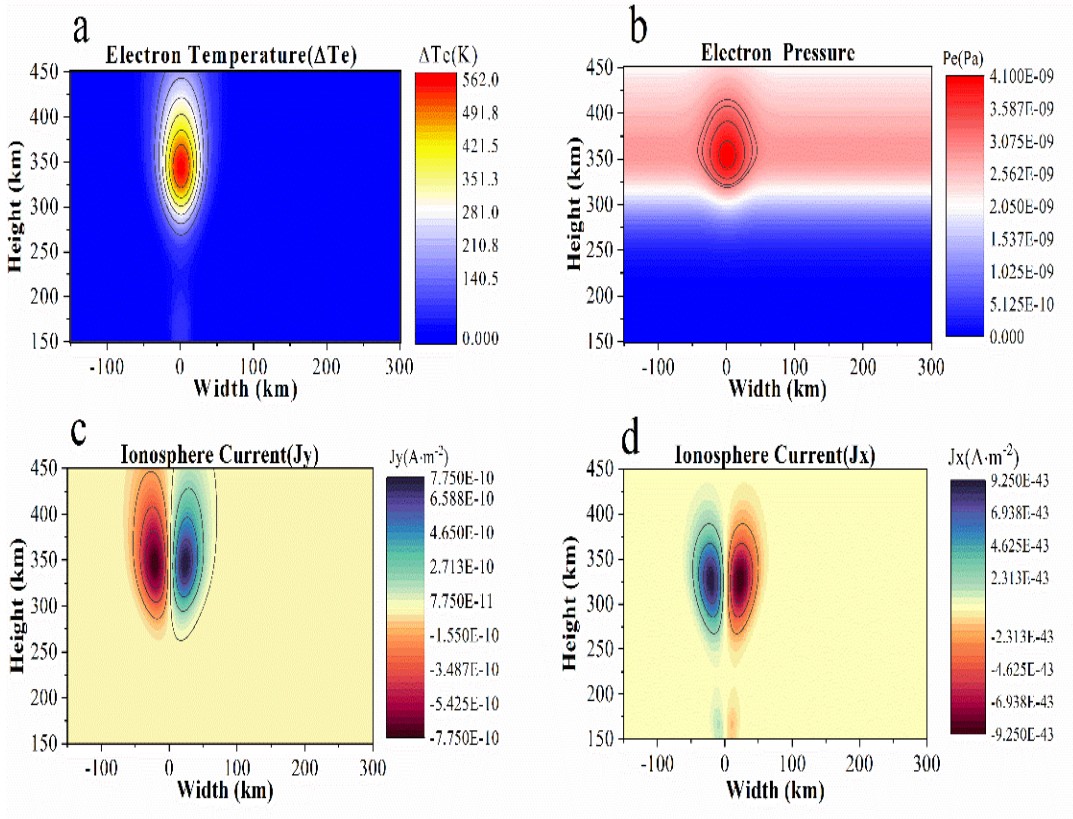

**Figure 2. Ionospheric parameter when the heating is stable. (a) Electron temperature. (b) Electron pressure, (c) $J_y$ current distribution.**
**(d) $J_x$ current distribution.**

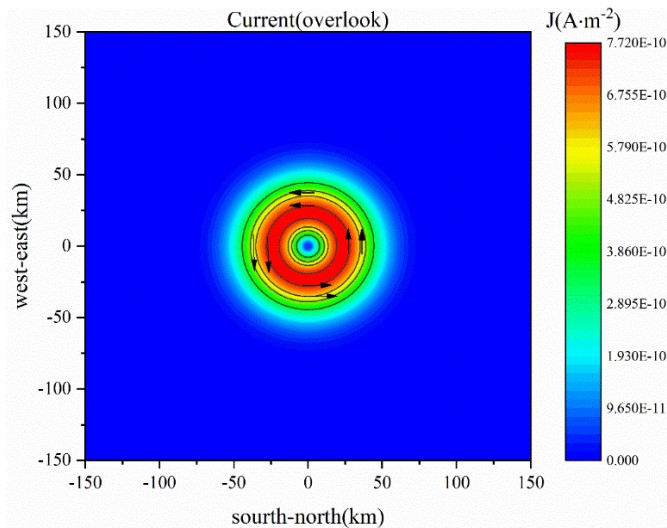

**Figure 3. Horizontal distribution of drift current at the critical frequency position.**

### 3.2 Influence of different angles on drift current

According to Kotik's experimental results, the strongest low-frequency electromagnetic signal is received on the ground when the HF wave heating direction is parallel to the magnetic field, i.e., the direction of the magnetic zenith. The radiated signal decreases as the angle between the radio wave and the geomagnetic field increases. This section provides a theoretical explanation for this observation. We study the effect of different heating directions on drift current by fixing other transmitter parameters and setting the angle $\theta = 10°, 20°, 30°$. The temperature change $\Delta T_e$ and current $J_y$ in the ionosphere are shown in Fig. 4. Figures 4(a), (c), and (e) show diagrams of electron temperature $\Delta T_e$ at $\theta = 10°, 20°, 30°$ respectively. It is obvious that with an increase of $\theta$, the heating area shifts horizontally, and the heating effect gradually weakens. Figure 4(b), (d), and (f) show diagrams of current at $\theta = 10°, 20°, 30°$. The currents undergo the same kind of change as the temperature. The current is symmetric about the launching center axis generally.

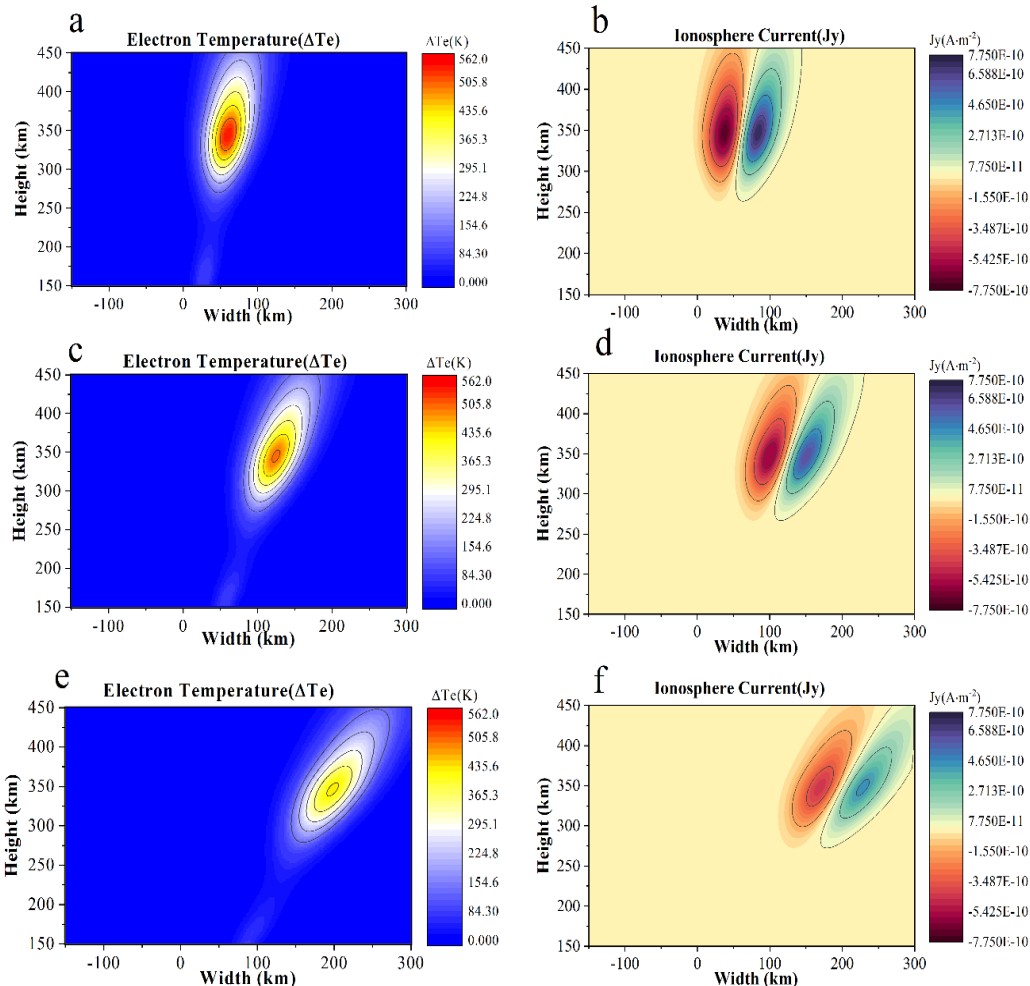

**Figure 4. Distributions of (a) electron temperature at $\theta = 10°$, (b) current $J_y$ at $\theta = 10°$, (c) electron temperature at $\theta = 20°$, (d) current $J_y$ at $\theta = 20°$, (e) electron temperature at $\theta = 30°$, (f) current $J_y$ at $\theta = 30°$.**

In order to investigate the effects of angle $\theta$ more visually, We calculate the maximum temperature change for different angle $\theta$; it is shown by the red dots in Fig. 5. The electron temperature change is 560 K at $\theta = 0$ and is reduced to 430 K at $\theta = 30°$. We also performed a plane integration of the absolute values of the current density (avoiding positive and negative cancellation); the results are marked by the green triangles in Fig. 5. It can be seen that the current reaches 5.76 A during vertical heating and decreases gradually with increasing angle.

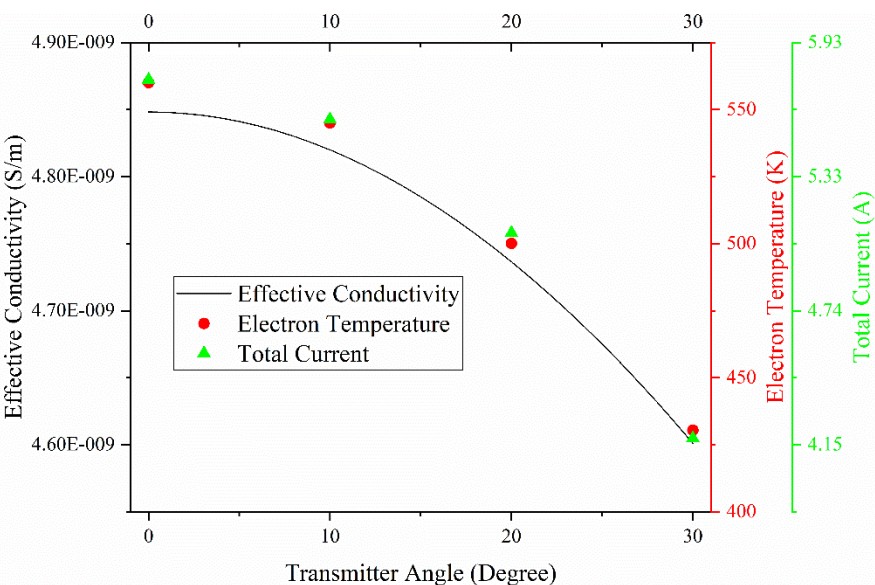

**Figure 5. Effective conductivity, electron temperature, and total current as functions of angle $\theta$.**

To explore what causes the changes of electron temperature and current, we calculate the effective conductivity at different angles. Combining Eq. (4) and (6), the dependence of effective conductivity on angle can be derived:

$$\sigma_{ef} = \frac{\varepsilon_0 \omega_{pe}^2 v_e}{\omega^2 + v_e^2} \sin^2(\theta) + \frac{\varepsilon_0 \omega_{pe}^2 v_e}{2} ((\frac{1}{(\omega - \omega_{ce})^2 + v_e^2} + \frac{1}{(\omega + \omega_{ce})^2 + v_e^2})(1 + \cos^2(\theta))$$
$$+ (\frac{1}{(\omega - \omega_{ce})^2 + v_e^2} - \frac{1}{(\omega + \omega_{ce})^2 + v_e^2}) \cos(\theta)) \tag{19}$$

Choosing the ionospheric frequency, electron cyclotron frequency, collision frequency, and transmitter frequency at the corresponding heights, we obtain a relationship between the angle and the effective conductivity, as shown by the black line in Fig. 5. It can be seen from the graph that the effective conductivity decreases gradually as the angle $\theta$ increases. We find that the trend of effective conductivity is the same as the trends of temperature and current. Physically, it is the change of effective conductivity that causes the change of heating. The conductivity is max when $\theta = 0$, where the heating effect is best and the current is the greatest. The conclusion could provide a natural explanation for the signal reaching its maximum values when the beam is directed along the Earth's magnetic field in Kotik's experiment (Kotik et al., 2013).

### 3.3 Magnetic field variations in the heating area

Unlike the methodology employed by Papadopoulos for measuring the ground magnetic signals excited by antimagnetic currents, we present computed results of magnetic signals in the heating ionospheric region, building upon the work of Lühr and Manoj, who utilized satellites to investigate the equatorial electrojet(Luhr et al., 2004; Manoj et al., 2006). In MHD, the

momentum equation is

$$\rho \frac{d\mathbf{u}}{dt} = -\nabla P + \mathbf{j} \times \mathbf{B} \tag{20}$$

where $\rho$ is mass density, $\mathbf{u}$ is fluid mass velocity, $\mathbf{j}$ is the electric current density, $p$ is pressure. During heating, the pressure variation is mainly contributed by electrons, so we ignore the ionic pressure and only consider the electron pressure. In this paper, we mainly consider the steady state. Inserting Maxwell's equation $\nabla \times \mathbf{B} = \mu_0 \mathbf{j}$, we get

$$\nabla \left( P_e + \frac{B^2}{2\mu_0} \right) = \frac{1}{\mu_0} (\mathbf{B} \cdot \nabla) \mathbf{B} \tag{21}$$

The right hand side represents the magnetic tension due to the curvature of the field lines; it is negligible since the scale of the heated region is very small compared to the scale of the geomagnetic field(Alken et al., 2017). Thus, electron pressure will immediately be balanced by a decrease in magnetic pressure

$$\delta P_e + \frac{B_1^{\,2}}{2\mu_0} = \frac{B_0^{\,2}}{2\mu_0} \tag{22}$$

where $B_0$ is the undisturbed geomagnetic field, $B_1$ is the disturbed geomagnetic field, $\delta P_e$ is the variation of the electron pressure due to heating. Thus the change of magnetic signal $\delta \mathrm{B} = B_0 - B_1$ induced by heating the ionosphere can be expressed as

$$\delta \mathrm{B} = B_0 - \sqrt{B_0^{\,2} - 2\mu_0 P_e} \tag{23}$$

Combining with the variation of electron pressure at $\theta = 0, 10°, 20°, 30°$, we can get the corresponding magnetic field variation $\delta \mathrm{B}$ in the heating region; the result is shown in Fig 6. As shown in Fig 6, it is clear that the magnetic field $\delta \mathrm{B}$ gradually decreases as the angle $\theta$ increases, which varies in the same way as the equivalent conductivity varies with angle. When $\theta = 0, 10°, 20°, 30°$, the maximum values of the magnetic field $\delta \mathrm{B}=48\mathrm{pT},47\mathrm{pT},43\mathrm{pT},37\mathrm{pT}$. Comparing the experimental results of Papadopoulos et al.( Maximum at the earth's surface is only about 1pT), one can find that the magnetic field in the heating region is much stronger than the strength of the magnetic field received at the ground surface, which indicates that the signal attenuates severely during the propagation process.

In this subsection, we calculated the magnetic field variation in the heating region based on the MHD theory. It is important to note that this calculation is only suitable for the heating region. A detailed calculation by propagation theory is needed for receiving signals at a distance (ground). This paper's model is based on the background conditions in high latitudes. Extending to the middle and low latitudes requires a similar transformation of the conductivity tensor. Hence, this model is not applicable to the middle and low latitudes.

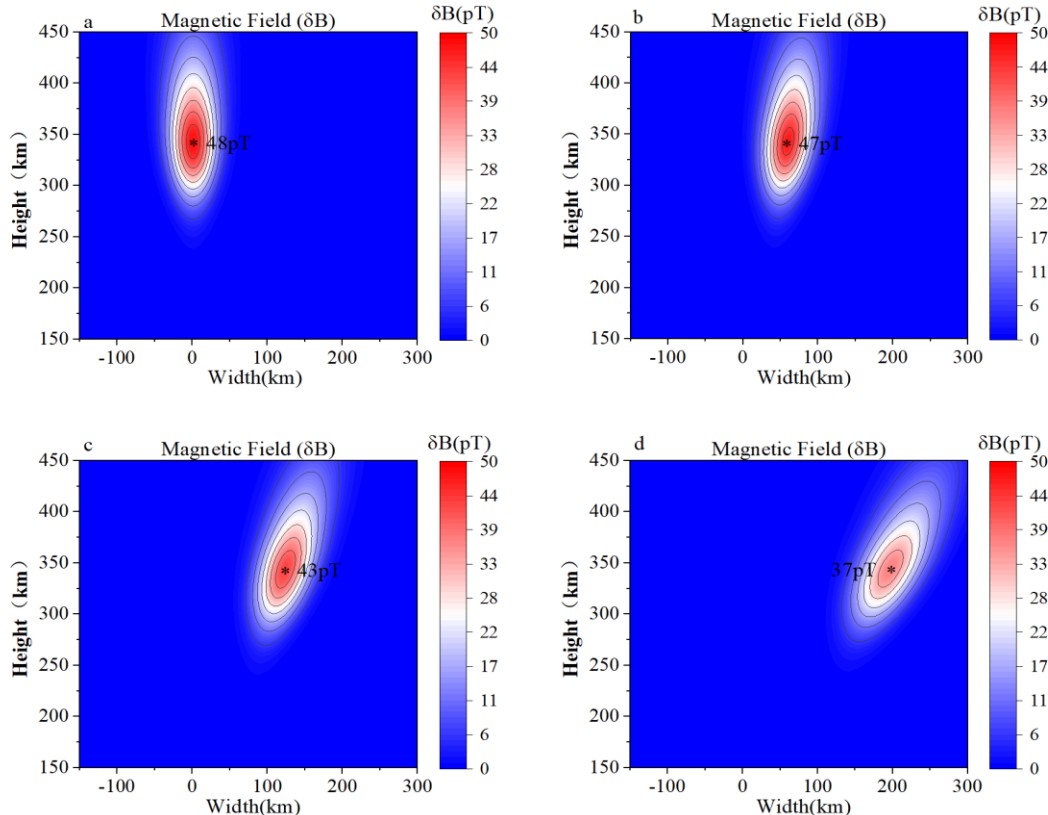

**Figure 6. Distributions of (a) magnetic field at** $\theta = 0$**, (b) magnetic field at** $\theta = 10^\circ$**, (c) magnetic field at** $\theta = 20^\circ$**, (d) magnetic field at** $\theta = 30^\circ$**.**

## 4 Conclusions

We establish a model of drift current in the ionosphere using the ohmic heating model and MHD momentum equation, and give the formulas to calculate the drift current and magnetic field variations in the heating area. The following conclusions are reached based on these calculations.

1) When the ERP is 500 MW and θ = 0°, the ionospheric electron temperature change $\Delta T_e$ is about 570 K, and the change ratio $\Delta T_e / T_e \sim 48\%$. From the calculated distribution of drift current in the ionosphere, the maximum value of Jy is approximately 7.8×10-10 A·m⁻², and Jx is approximately 9.3×10-43 A·m⁻². The total current excited by heating is 5.76 A.

2) It is concluded that the collisions of charged particles with neutral particles have a negligible effect on the current; electron-ion collisions do not affect the drift current. The current is mainly diamagnetic current and it is ring-shaped, zero at the center and gradually increasing outward until it decreases again.

3) An analytical equation of the dependence of effective conductivity $\sigma_{ef}$ on emission angle $\theta$ is given. The effect of the emission angle $\theta$ on the electron temperature and current density is explained by using the concept of effective conductivity. The most substantial current is obtained when X wave heats the ionosphere along the magnetic field direction, and the current gradually decreases as the angle $\theta$ increases. Theoretically, this explains why the strongest signal is received by the ground when heated along the magnetic inclination angle.

4) We give an equation for the magnetic field variation in the heating region. The calculation results show that the emission angle $\theta = 0, 10^\circ, 20^\circ, 30^\circ$, the maximum value of the magnetic field $\delta B = 48pT, 47pT, 43pT, 37pT$. The position of the maximum variation of the magnetic field is at the center of the heating area.

## Data availability

The ionospheric background parameters of our numerical simulation in this paper are fromhttps://kauai.ccmc.gsfc.nasa.gov/instantrun/iri/, and the atmospheric background profiles are from https://kauai.ccmc.gsfc.nasa.gov /instantrun/nrlmsis/.

## Author contributions

YL performed the modeling calculations and writing. HL and ZZX provided theoretical guidance, JW,XBL,CLand CXY
participated in the discussion and given valuable comments.

## Competing interests

The contact author has declared that none of the authors has any competing interests.

## Acknowledgments

This work is supported by " the Fundamental Research Funds for the Central Universities" (Grant No. HIT.OCEF.2022036)

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
