# Peer review of "F-region drift current and magnetic perturbation distribution by X wave heating ionosphere"

_EGUsphere, 2023_

## Author Response (AR1)

Dear Editors and Reviewers:

Thank you for your letter and for the reviewers' comments concerning our manuscript entitled " F-region drift current and magnetic perturbation distribution by X wave heating ionosphere." We are very grateful for the work you've done on this manuscript. Your comments are valuable and helpful for revising and improving our paper and providing important guidance for our research. We have studied the comments carefully and have made corrections, which we hope meet with approval. Revisions have been made according to the comments and the details of the revision are summarized in the following, with the comments in "black" and our responses in "blue":

**Referee #2**

1. In the abstract, the authors mentioned the following statement which is not appropriate. "the model explains why the radiation efficiency in Kotik's experiment was strongest 15 when the X wave is heating along the magnetic dip angle." You may revise the sentence as follow "the model explains why the radiation efficiency was strongest when the X wave is heating along the magnetic dip angle as reported in recent observations by Kotik et al….."

   Re: Thank you for the correction. We have modified it in the abstract.

2. There are many grammatical mistakes and very long sentences that make it difficult for the reader to follow. As an example please lines 30-34 and many more examples to be find.

   Re: Thanks to your comments, we have checked the manuscript and corrected long difficult sentences and grammatical problems in the text. Modification traces can be seen in the track-changes file.

3. Lines 44-46 have to be revised. This is not an appropriate way to promote your work. You may mention that your work is including more physics that was neglected in the previous studies.

   Re: Your comment do mean a lot to this manuscript. It is indeed inappropriate to represent one's work in this way in a manuscript. We rewrite this as follows:

   At present, theoretical research on ICD theory focuses mainly on the propagation process of ELF wave. In this paper, considering the effect of transmitter parameters and ionospheric parameters, we develop the ionospheric drift current and magnetic perturbation model by coupling the ohmic heating and MHD momentum equations. We then study the drift-current properties and the effects of collisions and transmitter angle on the drift current, and we calculate the magnetic field variation in the heating region.

4. Another issue that is not discussed or addressed properly here is the work done by Mahmoudian and Kalaee" Study of ULF-VLF wave propagation in the near-Earth environment for earthquake prediction, Advances in Space Research, Volume 63, Issue 12, 2019".This new study which is a continuation of previous work by Eliasson and Papadapoulos 2012, has shown that the VLF signal may not penetrate to the D-region as efficient as the ELF signal. I suggest that authors provide the explanation on ELF or VLF excitation during HF modulation (Beat wave) and try to provide a more general conclusion rather than just studying the excited diamagnetic current and its aspect angle dependency.

**Re:** Your comments do mean a lot to this manuscript. The propagation issues raised by the reviewers are essential. In the recommended literature, the authors demonstrate that " the efficiency of the induced currents in the ionosphere such as penetration is greatly limited as the transmission frequency," This work is significant. We will include a reference to it in the Introduction. However, in this paper, we are mainly concerned with the properties of the heating region when electromagnetic waves interact with the ionosphere. Therefore, we used an alternative method to calculate the magnetic field perturbation in the heating region. The specific procedure is as follows(from the revised manuscript):

Unlike the methodology employed by Eliasson and Papadopoulos for measuring the ground magnetic signals excited by antimagnetic currents, we present computed results of magnetic signals in the heating ionospheric region, building upon the work of Lühr and Manoj, who utilized satellites to investigate the equatorial electrojet(Luhr et al., 2004; Manoj et al., 2006). In MHD, the momentum equation is

$$\rho \frac{d\mathbf{u}}{dt} = -\nabla P + \mathbf{j} \times \mathbf{B} \tag{1}$$

where $\rho$ is mass density, $\mathbf{u}$ is fluid mass velocity, $\mathbf{j}$ is the electric current density, $P$ is pressure. During heating, the pressure variation is mainly contributed by electrons, so we ignore the ionic pressure and only consider the electron pressure. In this paper, we mainly consider the steady state.

Inserting Maxwell's equation $\nabla \times \mathbf{B} = \mu_0 \mathbf{j}$ , we get

$$\nabla \left( P_e + \frac{B^2}{2\mu_0} \right) = \frac{1}{\mu_0} (\mathbf{B} \cdot \nabla) \mathbf{B} \tag{2}$$

The right hand side represents the magnetic tension due to the curvature of the field lines; it is negligible since the scale of the heated region is very small compared to the scale of the geomagnetic field(Alken et al., 2017). Thus, electron pressure will immediately be balanced by a decrease in magnetic pressure

$$\delta P_e + \frac{B_1^2}{2\mu_0} = \frac{B_0^2}{2\mu_0} \tag{3}$$

where $B_0$ is the undisturbed geomagnetic field, $B_1$ is the disturbed geomagnetic field, $\delta P_e$ is the variation of the electron pressure due to heating. Thus the change of magnetic signal $\delta \mathrm{B} = B_0 - B_1$ induced by heating the ionosphere can be expressed as

$$\delta \mathrm{B} = B_0 - \sqrt{B_0^2 - 2\mu_0 P_e} \tag{4}$$

Combining with the variation of electron pressure at $\theta = 0, 10°, 20°, 30°$, we can get the corresponding magnetic field variation $\delta \mathrm{B}$ in the heating region; the result is shown in Fig 6. As shown in Fig 6, it is clear that the magnetic field $\delta \mathrm{B}$ gradually decreases as the angle $\theta$ increases, which varies in the same way as the equivalent conductivity varies with angle. When $\theta = 0, 10°, 20°, 30°$, the maximum values of the magnetic field $\delta \mathrm{B} = 48pT, 47pT, 43pT, 37pT$. Comparing the experimental results of Papadopoulos et al.( Maximum at the earth's surface is only about 1pT), one can find that the magnetic field in the heating region is much stronger than the strength of the magnetic field

received at the ground surface, which indicates that the signal attenuates severely during the propagation process.

In this subsection, we calculated the magnetic field variation in the heating region based on the MHD theory. It is important to note that this calculation is only suitable for the heating region. A detailed calculation by propagation theory is needed for receiving signals at a distance (ground).

[Figure]

Figure 6. Distributions of (a) magnetic field at $\theta = 0$, (b) magnetic field at $\theta = 10°$, (c) magnetic field at $\theta = 20°$, (d) magnetic field at $\theta = 30°$.

5.  It is critical that authors include the geomagnetic angles for other latitudes such as mid latitude (Arecibo Observatory) or equatorial region (possible facility coming at Hicamarca Peru). The efficiency of HF pump heating at those latitudes and possible variation of excited diamagnetic current should be discussed. If the proposed model is not capable of simulating such condition this weakness should be admitted in the text.

    **Re:** Thank you for the correction. After carefully considering the reviewers' comments, we realized that the conductivity tensor needs to be similarly transformed, and the current distribution should change when extending the model to low and middle latitudes. Our existing model is missing the corresponding part, so we declare the weakness of the model in the manuscript. The following description is made in our paper:

This paper's model is based on the background conditions in high latitudes. Extending to the middle and low latitudes requires a similar transformation of the conductivity tensor. Hence, this model is not applicable to the middle and low latitudes.

**Referee #2:**

1. This paper is a theoretical and numerical study of drift current model in the ionosphere by incorporating the ohmic heating model and the MHD momentum equation. The authors found that the ionospheric currents are mainly driven by a diamagnetic drift effect due to the ionospheric heating. The intensity of the ionospheric currents tended to decrease with an increase of the propagation direction angle of X wave. The main reason is that the effective conductivity becomes small for the large propagation angle. This result is clear and interesting, and I recommend that this paper should be published for Ann. Geophys. journal after some minor revisions. The revision points are shown below.

   **Re:** Thanks for your comments.Your acknowledgment of our work is an excellent incentive for us to keep going.

2. Line 55: The authors assume that the magnetic field inclination is 90 degrees for the calculation of X-wave heating. However, the magnetic field line has an inclination that is different from 90 degrees except for the magnetic pole. The authors should explain the validity of this assumption in this paper.

   **Re:** Thanks for your comments. There are two reasons why we assume a magnetic inclination angle of 90°.

   First, the location we chose for our study is HAARP. A query reveals that the magnetic inclination of the place is 75°, which is already relatively close to the magnetic inclination of the magnetic pole. Therefore, the magnetic inclination angle always defaults to 90° in previous studies, and this paper adopts the previous literature setting.

   Second, it can be found that the current (or effective conductivity) depends mainly on the angle between the magnetic inclination and the emission angle; the effect of the initial value is not so significant. Therefore, to simplify the calculation, we simplify the magnetic inclination to 90 °

3. Equation (9): Please replace ERP with one character. In this case, the reader may regard ERP as ExRxP.

   **Re:** Thanks for your comment. ERP is the effective radiated power of the transmitter. To avoid ambiguity, we rewrite *ERP* in Eq. 9 as $P_{ER}$.

4. Line 91: ...the electric force should be ...The electric force...

   **Re:** Thanks for your comment. we have made corrections in the manuscript.

5. Lines 105-107: Please include the references for each model. Further, the authors should show the version of the IGRF model used in this study. Please spell out IGRF.

**Re:** Thank you for your comments. According to the comments, the section was rewritten as follows:

Background data were acquired at HAARP on 2 October 2011. The ionospheric and atmospheric background profiles are given by the International Reference Ionosphere (IRI-2016)(Bilitza et al., 2022) model and the neutral atmosphere model (MSIS)(Picone et al., 2002), as well as geomagnetic field data from the International Geomagnetic Reference Field(IGRF-11) model(Geomagnetism et al., 2010).

**The references:**

Bilitza, D., Altadill, D., Truhlik, V., Shubin, V., Galkin, I., Reinisch, B., and Huang, X.: International Reference Ionosphere 2016: From ionospheric climate to real-time weather predictions, Space Weather, 15, 418-429, https://doi.org/10.1002/2016SW001593, 2017.

Picone, J. M., Hedin, A. E., Drob, D. P., and Aikin, A. C.: NRLMSISE-00 empirical model of the atmosphere: Statistical comparisons and scientific issues, Journal of Geophysical Research: Space Physics, 107, SIA 15-11-SIA 15-16, https://doi.org/10.1029/2002JA009430, 2002.

Finlay, C. C., Maus, S., Beggan, C. D., Bondar, T. N., Chambodut, A., Chernova, T. A., Chulliat, A., Golovkov, V. P., Hamilton, B., Hamoudi, M., Holme, R., Hulot, G., Kuang, W., Langlais, B., Lesur, V., Lowes, F. J., Lühr, H., Macmillan, S., Mandea, M., McLean, S., Manoj, C., Menvielle, M., Michaelis, I., Olsen, N., Rauberg, J., Rother, M., Sabaka, T. J., Tangborn, A., Tøffner-Clausen, L., Thébault, E., Thomson, A. W. P., Wardinski, I., Wei, Z., and Zvereva, T. I.: International Geomagnetic Reference Field: the eleventh generation, Geophysical Journal International, 183, 1216-1230, 10.1111/j.1365-246X.2010.04804.x, 2010.

**Dr. Lee**

I think author may add simulated results for other latitudes and make a comparison for them to enhance the practicality of this manuscript.

**Re:** Thank you for your comment. This paper's model is based on the background conditions in high latitudes. Extending to the middle and low latitudes requires a similar transformation of the conductivity tensor. Hence, this model is not applicable to the middle and low latitudes. However, we would happily devote some future work to this issue.